# Unsupervised Anomaly Detection for Cars CAN Sensors Time Series Using Small Recurrent and Convolutional Neural Networks

**DOI:** 10.3390/s23115013

**Published:** 2023-05-23

**Authors:** Yann Cherdo, Benoit Miramond, Alain Pegatoquet, Alain Vallauri

**Affiliations:** 1Renault Software Labs, 2600 Route des Crêtes, Sophia Antipolis, 06560 Valbonne, France; yann.cherdo@univ-cotedazur.fr (Y.C.);; 2LEAT (CNRS), Bât. Forum, Campus SophiaTech 930 Route des Colles, 06903 Sophia Antipolis, France

**Keywords:** anomaly detection, sensors, Internet of Things, unsupervised, Controller Area Network bus, car, time series, recurrent neural network, long short-term memory, gated recurrent unit, convolutional neural network, computational costs, anomaly likelihood

## Abstract

Predictive maintenance in the car industry is an active field of research for machine learning and anomaly detection. The capability of cars to produce time series data from sensors is growing as the car industry is heading towards more connected and electric vehicles. Unsupervised anomaly detectors are therefore very adapted to process those complex multidimensional time series and highlight abnormal behaviors. We propose to use recurrent and convolutional neural networks based on unsupervised anomaly detectors with simple architectures on real, multidimensional time series generated by the car sensors and extracted from the Controller Area Network bus (CAN). Our method is then evaluated through known specific anomalies. As the computational costs of Machine Learning algorithms are a rising issue regarding embedded scenarios such as car anomaly detection, we also focus on creating anomaly detectors that are as small as possible. Using a state-of-the-art methodology incorporating a time series predictor and a prediction-error-based anomaly detector, we show that we can obtain roughly the same anomaly detection performance with smaller predictors, reducing parameters and calculations by up to 23% and 60%, respectively. Finally, we introduce a method to correlate variables with specific anomalies by using anomaly detector results and labels.

## 1. Introduction and Related Work

Predictive maintenance in the car industry refers to the anticipation and management of failures and abnormalities occurring in a car. For example, if the oil pressure remains too low for a long period of time, while it will not cause any immediate damage, it might induce motor and mechanical parts’ failures and breaks in the near future. The user will then have to go to the garage, broken parts will be replaced, and a root cause of the failure will be identified, if possible. Anticipating and understanding such a failure occurrence can bring about the following outcomes. First, comprehension of the root causes of part failures allows for better design and manufacturing, enhancing the generation quality of future products. Second, anticipating a part failure allows for better purchase management and logistics for the replacement parts, resulting in financial optimization. Finally, it helps the garage to better solve the issue by providing more information and knowledge about the failure.

One can comprehend a vehicle as a complex electric and mechanical system that has an inner control and sensing system. In Section 2, we offer more detailed information about car architecture. The data produced by sensors, for example the oil pressure or the battery voltage, can be extracted and then processed off-board. This results in multidimensional time series data. As the car industry progresses towards more connected, electrical, and complex vehicles, the amount of data produced by sensors that can be processed for predictive maintenance grows both in quality and quantity.

While ML algorithms are being used to process such information on the cloud, two main costs are being generated: the infrastructure cost due to algorithm computations and the bandwidth cost due to data transfers from cars to the cloud. Bandwidth costs and data storage capacities can be blocking constraints in certain cases. Personal data protection can also be problematic when trying to store personal cars’ usage data in the cloud. For all of these reasons, getting the ML algorithms closer to the data producer, namely the car, as an embedded system is a rising research topic. The main constraint is the low computational capacity that is generally available in a vehicle. In this paper, we address this issue in the context of unsupervised anomaly detection by lowering the size of the ML algorithm while maintaining the same detection performance. The algorithm itself being a source of complexity, we chose the simplest neural networks from among the best time series anomaly detectors in the literature.

Building supervised machine learning algorithms to predict failures from sensors’ time series would be ideal. However, obtaining the labels of failures is not so straightforward and, in certain cases, can introduce significant noise. In fact, failures are not directly stored as a statistic, but instead, one can record parts that have been replaced at the garage and try to predict parts which will need to be replaced. When a vehicle is handled in a garage because of a failure, replaced parts are not always linked to the real failure due to expertise error and habits. Therefore, only a subset of possible failures can be covered by this kind of approach. In practice, datasets of parts’ replacements tended to be heavily unbalanced.

On the other hand, the production of failures labels by experts would be very expensive considering the complexity of parts and car models’ and scenarios’ variations. Unsupervised anomaly detection could therefore be considered as a solution to this problem.

Anomaly Detection is a wide research field with many different techniques [1,2]. Unsupervised anomaly detection refers to the problem of finding abnormal patterns in the data. In order to apply unsupervised anomaly detection within time series, companies such as Amazon, Twitter, Etsy, and Yahoo have developed their own models that are a mixture of classical statistical, decomposition, and machine learning algorithms [3,4,5,6]. Some models are also bio-inspired, e.g., modeling the episodic memory of the cortex [7] using a model called hierarchical temporal memory (HTM) [8]. Another bio-inspired model has been presented in Reference [9]. In this work, an evolving spiking neural network (OeSNN) is trained similarly to the HTM to learn sequences continuously. Both HTM and OeSNN detect two kinds of anomalies. The first one is detected when the model predicts the wrong value, and the second one is detected when the model is unable to predict anything from the given context. Those models use local learning rules like Hebbian or genetic rules that have been shown to be less powerful than the well-established gradient descent optimization. In particular, these methods suffer from critical hyper parameters’ selection.

Convolutional neural networks (CNN) have been used to efficiently learn normal patterns and reveal anomalous ones [10]. This model was also successfully applied to spectral residual in Reference [11]. The field of graph theory also proposed a graph- and attention-based anomaly detector for multivariate time series in Reference [12].

Heavier models like generative adversarial networks (GAN) based on bidirectional Long Short Term Memory (LSTM) were proposed in Reference [13]. Modeling such correlations over time, or building a temporal model of time series, is an active area of development with manifold solutions [14]. One of the state-of-the-art approaches is LSTM [15]-based anomaly detection, which has had a great impact on the field of unsupervised anomaly detection [16]. In Reference [17], LSTM neural networks were used to predict time series after training on normal time series without any anomalies. A threshold was then applied to the error between the learned prediction and the true time series in order to find anomalous patterns. Some simple one-layer LSTM networks can be used as in Reference [16], although stacked (or deep) LSTM networks with several layers show better results, as demonstrated in References [17,18,19,20,21]. Gated recurrent unit (GRU) [22] is a lighter version of LSTM also in the family of recurrent neural networks (RNN), and it shows very similar performance in time series modeling. We propose to compare LSTM and GRU models in terms of detection performance and computational costs in order to further our goal of lowering the computational cost of our anomaly detector.

In the car industry, HTM have been used to conduct unsupervised anomaly detection on time series using nine variables provided by racing cars’ recordings [8]. Several papers can be found about intrusion detection, safety, and cybersecurity, as connected cars are exposed to hacks [23,24,25]. Intrusion detection conducted using a complex CNN-LSTM and attention mechanisms in CAN traffic logs can be found in Reference [26]. Within the same application, a multilayer perceptron was employed in Reference [27]. These works focus on the way information transits through the CAN and not on the information itself, as this is sufficient to detect an intrusion. Fewer works have been found that focus directly on the information provided by the car sensors and the commands that transit through the CAN, such as oil pressure, engine temperature, etc. This approach can be used for intrusion detection but also allows for the detection of abnormal complex behavior happening inside the car. In Reference [28], a more classical and statistical method, the Kalman filter, is employed to model such time series and to detect any that are abnormal. In the same context, hidden Markov models have been used to find abnormal internal behaviors [29]. For this kind of car sensor data, we propose the use of LSTM-, GRU- and CNN-based anomaly detectors.

One can find a large community of researchers working on image and video classification for autonomous driving cars. This field also includes anomaly detection, as is exhaustively presented in Reference [30]. Finally, an original approach that uses acoustic recordings inside the car to find anomalies is proposed using feature extractions and deep autoencoders in Reference [31]. It can be seen that most of the state-of-the-art unsupervised anomaly detection algorithms have yet not been applied to car sensors’ time series. The goal of this paper is to prove that such a task can be achieved using small yet powerful anomaly detectors. This is why, among all available models, LSTM, GRU, and CNN were chosen for use. These neural networks allow architectures to be kept simple while using the powerful gradient descent optimization method.

In this paper, our contribution is to apply state-of-the-art LSTM- and CNN-based, unsupervised anomaly detection to real car time series and to evaluate the results with respect to abnormal labels. We do not focus on any specific kind of anomaly, and our solution aims at highlighting any complex multidimensional behavior of a vehicle. We then compare our models with different amounts of layers and cells and show that similar anomaly detection performance can be obtained using smaller models even if this limits their prediction accuracy. We finally introduce a method of correlating variables to those specific anomalies using the anomaly detector results and labels.

Our paper is organized as follows. In Section 2, we introduce the data that we are working on, focusing on how they have been produced and extracted from the vehicle and the workflow leading to anomaly detection. In Section 3, we describe in detail the addressed issue and the three state-of-the-art models and architectures we used, including the entire process of unsupervised anomaly detection. In Section 4, we specify the performance metrics we used to evaluate our models and the qualitative results we obtained from the aforementioned data. In this section, we also introduce and evaluate a method for finding variables that are the most correlated with the abnormal labels. In the last section, we discuss the overall work presented in this paper in terms of opportunities and limitations, focusing on our main contributions. We then introduce possible future work that could enhance our models based on our findings.

## 2. Car Time Series Extracted from the CAN Bus

In this section, we present the process of data generation in the car and how it can be exploited for machine learning purposes.

### 2.1. The Data and the Car

A car is a complex assembly of mechanical and electrical parts. The control and sensing of all of those parts are processed by electric computing units (ECU). ECUs are microcontroller-based units that can receive, process, and send information from and to other ECUs. Specifically, information transits through a controller area network (CAN) bus. For example, one ECU might control the oil temperature while receiving and processing information about the motor rotational speed that has been sent from the ECU managing the motor area. In a modern car, tens of ECUs can process information simultaneously while the car is running. Information transiting through the CAN bus is thus a very good reflection of the car’s global state. In order to extract this information, a spy system can listen to the information transiting on the CAN bus and save it in a structured manner, associating a sample to its variable and timestamp. Then, the resulting data can be sent to a cloud database and into an anomaly detector, as shown in Figure 1.

### 2.2. Our Dataset

The dataset used in this paper consists of the recordings of 486 variables gathered from an Alpine Renault car during driving tests conducted on circuits. Those tests were conducted over the course of 4 months, resulting in 17 GB of data. The sampling rate of the recordings was 10 Hz. In order to simplify the training and evaluation of the model, we filtered variables with respect to their complexity and retained only the richest ones. As many variables correspond to commands with only one to three discrete values and are rarely triggered, most of the signals show very few variations through time and are therefore poorly informative. Those are the variables which we filtered out. This process resulted in an 85-variable system. As recordings do not have the same temporal length, this dataset can be seen as a set of multivariate time series with different lengths. Finally, we resampled the data to 1 Hz using the average in order to save time on training and testing. This dataset is not public but is necessary in order to apply state-of-the-art anomaly detectors to real case studies. Nevertheless, we have included a comparison of different state-of-the-art models as well as the ones we have chosen in Table 1. These were applied to an open-source benchmark in order to ensure that our method and implementation were valid.

### 2.3. Labels

The utilized labels consist of 3683 timestamps in which an anomaly linked to the oil pressure was noticed. They were automatically generated following expert rules over five key variables, namely the oil pressure, the oil temperature, the mean effective torque, the engine RPM, and the engine coolant temperature. The abnormality is here defined as problematic oil pressure behavior. It is worth noting that this highly specific abnormal behavior does not cover all possible abnormal behaviors of the car and thus provides a limited evaluation of our unsupervised anomaly detector.

## 3. System Model and Problem Formulation

### 3.1. System Model

The overall system model is represented in Figure 2. We considered a time series T of n samples, each sample being an indexed vector of m dimensions representing variables with n∈N and m∈N, such that
(1)$$T=\{x_{0},x_{1},\ldots ,x_{n-1}\}.$$

Each time series sample $x_{t},t\in \left\{0,\ldots ,n-1\right\}$ is potentially corrupted by an anomaly that modifies its “normal behavior” in the sense that the value registered at this point does not fit with the usual pattern of the time series. The goal of the anomaly detector is then to associate each sample $x_{t}$ with an estimated binary anomaly label ${\hat{a}}_{t}\in {\left\{0,1\right\}}^{m}$ in order to indicate whether said sample is corrupted by an anomaly. These labels then form the time series A.
(2)$$A=\{{\hat{a}}_{0},{\hat{a}}_{1},\ldots ,{\hat{a}}_{n-1}\}.$$

### 3.2. Unsupervised Anomaly Detection

In this section, we review our baseline algorithm formed by a stacked LSTM-based anomaly detector from References [17,19]. We added the anomaly likelihood method to the anomaly detection process as proposed in Reference [7]. This method allows the anomaly detector to focus on dynamic variations of the prediction error. As different variables at different moments can present behaviors that are more or less difficult to predict, the error prediction average might vary, and this makes the usage of a simple threshold tricky. Anomaly likelihood adapts nicely to this complex behavior, as it evaluates the local probability of a prediction error to occur and also prevents random point anomalies. Finally, we explain below how we acquired all of the parameters.

In Table 1, we have gathered the anomaly detection results found in References [9,13] on the public Yahoo benchmark [32]. We added our LSTM with 50 cells and two layers, as presented in Section 4, using semi-supervised threshold optimization, just as in the work of Reference [10]. For this model, we used 100 epochs, a learning rate of $10^{-3}$, and a time window of w=100. We can see that LSTM and CNN performed well compared to some of the more complex models such as generative adversarial networks (TadGAN) or those that are bio-inspired. As we wanted to use small models with simple architectures in order to reduce computational costs while still providing good performance, we chose to exploit LSTM and CNN predictors in this paper. We also chose to compare the GRU [22] with the LSTM, as it is a simplified version of the LSTM. All of these predictors are described below.

### 3.3. LSTM Predictor

The first block of the anomaly detector is formed by an LSTM neural network, which is an RNN that has proven to be very efficient in capturing the temporal dependencies of a time series using internal memory [15]. The LSTM cell is described in Figure 3. Each cell is recurrent and outputs to vectors $c_{t}$ and $h_{t}$. One LSTM layer can be made of several cells. In the following sections, we use the notation LSTM (n,m) to refer to an LSTM with *x* cells in the first layer and *y* cells in the second layer.

The overall stacked LSTM we used has two layers. It takes a sample $x_{t}$ as an input and predicts the next sample $x_{t+1}$ while using its internal state recurrently. A simple linear layer allows for the conversion of the LSTM output vector $h_{t}$ into a prediction of $x_{t+1}$, as shown in Figure 4.

In this paper, we also compare the LSTM with two other models.

### 3.4. CNN Predictor

A convolutional neural network (CNN) is a well-known machine learning algorithm that has been proven to be very powerful, especially for image classification. In Reference [10], it was used to predict time series and to detect anomalies the same way we proposed in this paper. It is one of the state-of-the-art approaches to unsupervised anomaly detection. The limitation of a CNN lies in its inability to integrate temporal information longer than its convolutional window. As can be seen in Figure 5, the time series is cut into windows in which we run the convolutions. The model can then predict the next time series sample $x_{t+w+1}$ given the window $\left\{x_{t},\ldots ,x_{t+w}\right\}$. This CNN model has two layers made up of a 1D convolution layer followed by an ReLU activation function. $l_{t}$ and $h_{t}$ are intermediate hidden vectors generated by the model. In the following sections, we use the notation CNN (n,m) to refer to a CNN with n channels in the first layer and m channels in the second layer.

### 3.5. GRU Predictor

Finally, we propose to use a gated recurrent unit (GRU) network [22], which is a simplified version of the LSTM with fewer gates in each cell forming a lighter RNN. Its architecture is exactly the same as that of the LSTM shown in Figure 4. Depending on the complexity of the task, a GRU can sometimes perform just as well or even better than an LSTM while having fewer parameters. We show such a result very clearly in Table 2. In the following sections, we use the notation GRU (n,m) to refer to a GRU with n cells in the first layer and m cells in the second layer.

### 3.6. Anomaly Detection

During training, for each sample $x_{t}$, the predictor output a prediction ${\hat{x}}_{t+1}$. The mean square error (MSE) over all dimensions and time was then computed between the predicted and true samples of one time series.
(3)$$\mathrm{MSE}=\frac{1}{\mathrm{nm}}\sum_{t=0}^{n-1}\left|\right|{\left({\hat{x}}_{t}-x_{t}\right)}^{2}{\left|\right|}_{1}$$
where ${\left||.|\right|}_{1}$ stands for the norm 1 of a vector.

This error was used as a loss function in order to optimize the predictor using a gradient descent optimization. Using this method and using an LSTM or a GRU, we were able to take full advantage of having time series of different lengths in the dataset. No information was lost due to padding or cutting the time series. The predictor was able to efficiently adapt to the variation in time series length.

In a vehicle, most of variables that we exploited here are correlated; for example, a failure in the oil pressure impacts the engine temperature. We made use of this assumption and provided all of the m variables to the predictor together in order to help it to capture more-complex temporal and cross-variable correlations and better predict the signal.

By comparing the true values of the *t*th sample $x_{t}$ with the predicted values, the prediction squared error for the *t*th sample was formed as:(4)$$e_{t}={\left({\hat{x}}_{t}-x_{t}\right)}^{2}$$

We then obtained the time series of the squared prediction errors:(5)$$E=\{e_{0},e_{1},\ldots ,e_{n-1}\}$$

We then applied the anomaly likelihood method to the time series E as in reference [7] so as to obtain the anomaly scores $s_{t}$ and labels ${\hat{a}}_{t}$. Note that we adapted its expression here for the use of multidimensional time series.

First, we computed the parameters $\mu_{t}$ and $\sigma_{t}^{2}$ of a sliding normal distribution of w samples of E.
(6)$$\mu_{t}=\frac{\sum_{i=0}^{w-1}e_{t-i}}{w}$$
(7)$$\sigma_{t}^{2}=\frac{\sum_{i=0}^{w-1}{\left(e_{t-i}-\mu_{t}\right)}^{2}}{w-1}$$

We then used the Gaussian tail probability (Q-function [34]) of the recent sliding average of $W^{\prime }$ samples of prediction errors in order to obtain the anomaly score.
(8)$$s_{t}=1-Q\left(\frac{\tilde{\mu_{t}}-\mu_{t}}{\sigma_{t}}\right)$$
where,
(9)$$\tilde{\mu_{t}}=\frac{\sum_{i=0}^{W^{\prime }-1}e_{t-i}}{W^{\prime }}$$

Finally, we used a threshold vector θ to compute the found anomalies.
(10)$${\hat{a}}_{t}^{i}=\left\{\begin{array}{cc} 1 & \;\mathrm{if}\;s_{t}^{i}\geq \theta^{i}\\ 0 & \;\mathrm{otherwise} \end{array}\right.$$
where ${\hat{a}}_{t}^{i}$ is the scalar of the *i*th dimension of the vector ${\hat{a}}_{t}$.

After the prediction, each variable was treated separately with its own parameters; thus, there were m anomaly likelihoods being processed in parallel.

We finally defined the set $A_{T}$ of timestamps in which at least one variable was found to be abnormal.
(11)$$A_{T}=\{t\in \left[t_{O},t_{n-1}\right]\;|\;||{\hat{a}}_{t}{\left|\right|}_{1}>0\}$$

With $t_{i}$ being the timestamp of the corresponding time series sample.

## 4. Results

After learning the predictor parameters, we ran our unsupervised anomaly detector through all the time series of our dataset and then exploited the results through different evaluations.

For all models, during the learning phase we used 300 epochs, a learning rate of $10^{-4}$ with the Adam optimization, and 2000 subsequences of 100 points each; additionally, we used 20% of the data for testing. The Adam optimization is one of the best and most-utilized gradient-descent-based optimization algorithms for neural networks such as CNN and RNN that exists in the field of machine learning [35].

Below, we will first evaluate how well our algorithm catches the given anomalies using our specific labels, and then we will propose measuring the correlation between labels and variables using the multidimensional nature of our algorithm.

### 4.1. Evaluation with Labels (Oil Pressure Failure)

As explained in Section 2, labels that have been provided to our team express a specific abnormal behavior due to failures in oil pressure and do not cover all possible abnormal behaviors.

These labels point to specific timestamps in the dataset where this oil pressure failure has been detected. This abnormal behavior is not instantaneous and appears for several seconds. Thus, we have to consider a time window $w_{a}$ around those timestamps’ labels to assess whether an abnormal sample found by our algorithm matches one label. We define the set of true positives TP as:(12)$$\mathrm{TP}=\{t\in A_{T}\;|\;\exists t_{l}\in L,\;t_{l}-w_{a}\leq t\leq t_{l}+w_{a}\}$$

With L and $A_{T}$ being, respectively, the set of labels and found anomalies timestamps.

We only consider one true positive sample with respect to one label even if there are several true positives in that label window, hence the usage of the set formalism.

We then define the false positive (FP), true negative (TN) and false negative (FN) sets as follows.
(13)$$\mathrm{FP}=\{t\in A_{T}\;|\;∄t_{l}\in L,\;t_{l}-w_{a}\leq t\leq t_{l}+w_{a}\}$$
(14)$$\mathrm{TN}=\left\{t\in \left[t_{O},t_{n-1}\right]\backslash A_{T}\;|\begin{array}{c} ∄t_{l}\in L,\\ \;t_{l}-w_{a}\leq t\leq t_{l}+w_{a} \end{array}\right\}$$
(15)$$\mathrm{FN}=\left\{t\in \left[t_{O},t_{n-1}\right]\backslash A_{T}\;|\begin{array}{c} \exists t_{l}\in L,\\ \;t_{l}-w_{a}\leq t\leq t_{l}+w_{a} \end{array}\right\}$$

In order to obtain a clearer evaluation, we then compute respective rates—namely the true positive rate (TPR) for TP and so on—corresponding to those last ensembles as:(16)$$\mathrm{TPR}=\frac{\left|\mathrm{TP}\right|}{\left|L\right|}$$
(17)$$\mathrm{FPR}=\frac{\left|\mathrm{FP}\right|}{n}$$
(18)$$\mathrm{TNR}=\frac{\left|\mathrm{TN}\right|}{n}$$
(19)$$\mathrm{FNR}=\frac{\left|\mathrm{FN}\right|}{n}$$

Provided in Table 3 are the TPR, FPR, TNR, and FNR obtained with windows $w_{a}$ of 5, 30, and 60 s and an LSTM with two layers of 50 cells with the notation LSTM (50-50).

The best results were obtained using the largest $w_{a}=60$ s, and they achieved 86% successful detection.

The FPR is 0.061%. If false positives were normally distributed over time and considering our 1 Hz sampling rate, this would mean that the model sends a false positive alert, on average, every 16 s. First, we have to remember here that our labels are limited to one specific kind of anomaly, and thus, this relatively high FPR might not depict an unsatisfactory behavior of our unsupervised anomaly detector. A subset of anomalies might correspond to actual abnormal behaviors of the vehicle which are not given in our labels.

The evaluation of whether our false positives match abnormal behaviors is not always straightforward. Visual interpretation can be difficult, as the model processes 85 variables altogether.

In Figure 6, we show an example of a successful detection made by our model regarding the oil pressure problem. As the engine mean effective torque is rising, the oil pressure should do the same; however, it does not do so here, which constitutes abnormal behavior. We also see how anomaly likelihood offers a nice dynamic adaptation of the quadratic error processing to find anomalies.

### 4.2. LSTM Computation Costs Comparison

Here we evaluate LSTM models with respect to different amounts of cells and layers. We start with our baseline LSTM (50-50) and run smaller architectures. We also compute the number of parameters and multiply–accumulate operations (MACs) for each model in order to evaluate their computational costs using the pytorch-OpCounter (access date: 12 January 2022) library https://github.com/Lyken17/pytorch-OpCounter.

In Table 4, the LSTM (10) shows the best TPR, but, for a better evaluation, the FPR has to be considered, as discussed below.

When evaluating the anomaly detection results, the two main factors to be considered are the TPR and the FPR. We want the highest TPR while keeping the FPR as low as possible. Depending on the threshold used during anomaly detection, TPR and FPR vary. It is difficult to equalize at least one of these across different models. Thus, we propose to use the positive likelihood ratio (PLR) defined in Equation (Equation 20), which allows for better evaluation and comparison between different models.
(20)$$\mathrm{PLR}=\frac{\mathrm{TPR}}{\mathrm{FPR}}$$

In Table 5, we first see that the LSTM (50) gives the best PLR of 14.2. We can also observe that the LSTM (10) with only 10 cells performs very similarly with a PLR of 14.1 but offers a computation cost about 6.6 times lower. As a result, our best predictor for anomaly detection is the LSTM (10). This is one of the lightest in terms of parameters and MACs, which makes it a better candidate for embedded systems. However, it is also one of the worst predictors with an MSE of 0.0059. This is an interesting property of the overall anomaly detector process that allows for the use of simpler predictors despite the prediction performance limitation.

In order to evaluate more directly and quantitatively each state-of-the-art model, including the CNN [10], we have tested all models with various sizes and without the anomaly likelihood. The method used to find anomalies this time is semi-supervised, and we also use the f1 score defined in Equation (Equation 23) to evaluate our results [10]. The model is processed exactly as before for training and prediction.
(21)$$\mathrm{recall}=\frac{\mathrm{TP}}{P}$$
where TP is the number of true positives and P the number of real positives.
(22)$$\mathrm{precision}=\frac{\mathrm{TP}}{\mathrm{PP}}$$
where PP is the number of predicted positives.
(23)$$f1=2\cdot \frac{\mathrm{precision}.\mathrm{recall}}{\mathrm{precision}+\mathrm{recall}}$$

Instead of using the anomaly likelihood method, we simply search for the anomaly threshold that maximizes the PLR. This threshold is the value above which the prediction error is considered to be abnormal. As finding this threshold requires labels, this method is therefore semi-supervised. This allows us to focus our performance analysis more on the ML model itself and avoid unsupervised post-processing bias.

In Table 2, all of the models’ performances are given in detail. We can see that the best f1 score was achieved by the CNN (32-32) followed very closely by the GRU (50-50) which is much lighter in terms of computational costs.

Considering the window $w_{a}=5$ s for example, we observe that the CNN (8-8), the GRU (50,50), and the LSTM (50,50) reach the same f1. However, this GRU has 23% reduced parameters and consumes 23% fewer MACs than this LSTM. This GRU, while having 3.79 times greater parameters than this CNN, still consumes 60% fewer MACs than this CNN. This result highlights that CNN might not always represent the best solution for time series, as convolutions induce a lot of calculation over a given time window. As no information is integrated from one time window to another, a CNN indeed needs to process *N* sliding time windows while an LSTM or a GRU processes *N* samples.

In terms of TPR and FPR, the GRU also shows the highest performance with the lowest computational costs. From one time window $w_{a}$ to another, the signal is being resampled in our implementation method, meaning that signals do not have the same length and some labels can be merged together. This has an impact over the range of f1, TPR, and FPR scalars, which makes comparisons between those metrics across time windows difficult.

In accordance with the previous results, we see that the f1 score, TPR, and FPR are not or very slightly impacted by the model size whereas the prediction performance increases accordingly to with size. We thus show again that the best prediction model might not be the best anomaly detection model in terms of anomaly detection performance and computational costs. Finally, the proposed semi-supervised anomaly threshold optimization method offers very low or null FPRs.

In the next section, we present a method to correlate variables with a specific kind of anomaly.

### 4.3. Correlation of Variables with Labels

As our labels are focused on one specific abnormal behavior, the question was raised as to whether it is possible to explain which variables are linked to that behavior.

In our case, abnormal behavior has been defined by experts using five expert variables: MeanEffTorque, EngineRPM, RST_EngineOilPressure, RST_EngineCoolantTemp, and RST_OilTemperature. The anomaly is linked to a problem with the oil pressure.

Making use of our anomaly detector and the way it handles each variable independently after prediction, we propose to use our model PLR to correlate variables with the anomaly, the underlying idea being that variables showing better detection are more linked to the anomaly.

We look at variables with the highest PLR. In Figure 7, we see that three of our five expert variables are ranked within the first quarter of variables with the best PLR out of the total 85 variables. Modifying the parameters of the anomaly detection model will impact this result, but we always find most of the five expert variables in that best quarter. Please note that we find oil pressure to be the second-best variable. This method thus effectively points out variables that are correlated with the anomaly, although we would need additional expert feedback in order to understand the role of the neighbor variables. We also want to know whether they are correlated to the anomaly. Unfortunately, we were not able to obtain that feedback yet.

## 5. Discussion and Future Work

In this paper, we have proposed unsupervised anomaly detection models using different state-of-the-art algorithms such as a LSTM, GRU, or CNN and anomaly likelihood or a semi-supervised anomaly threshold optimization. We applied these models in order to effectively point out abnormal temporal behaviors over a real, complex, multivariate sensor time series that was extracted from a vehicle CAN bus during circuit tests.

The evaluation of these models with respect to some limited labels shows an acceptable rate of detection, although there is still room for improvement. First, as our labels only express one specific abnormal behavior, it is difficult to fully evaluate our model. Second, further advanced data prepossessing and feature engineering conducted with experts would certainly help.

The comparison between different LSTMs, GRUs, and CNNs shows that the anomaly detector can work just as well or even better with smaller predictors despite a slight decrease prediction performance. Under certain circumstances, we have shown that the GRU can save up to 23% of parameters and 60% of MACs operations compared to the LSTM and the CNN. This is an interesting property for embedded systems, as it allows for a reduction of the algorithm computational cost.

As future works, we intend to embed anomaly detectors inside vehicles and thus expose our models directly to high computational cost constraints. We are currently studying spiking neural networks as a candidate to tackle this problem. An in-depth anomaly detection performance and computational costs study should be performed in the future on public data benchmarks so as to focus on the reduction of computational costs, as this is lacking in the literature presented in this paper.

Finally, we proposed a method for evaluating the correlation between variables and a specific abnormal behavior given in labels. This method shows interesting results and would need further in-depth evaluation in order to refine it. 

## Figures and Tables

**Figure 1 sensors-23-05013-f001:**
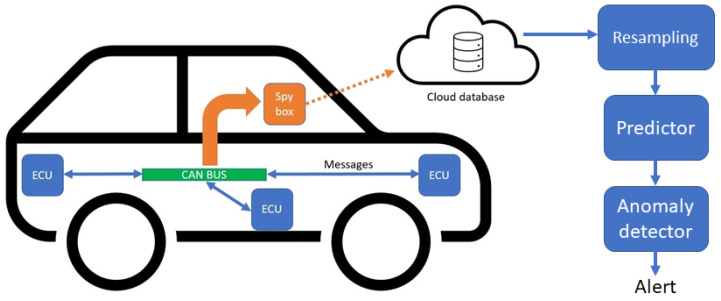
Flow chart presenting the process of data generation in a modern car leading to anomaly detection. Within the car, ECUs handle sensors and communicate some of their values through the CAN bus. That CAN bus can be read in order to extract those sensors’ data and can then be sent to the cloud or be exploited internally by another ECU. The resulting time series are then resampled, a prediction is emitted after each sample, and an anomaly is either detected or not given the prediction error.

**Figure 2 sensors-23-05013-f002:**
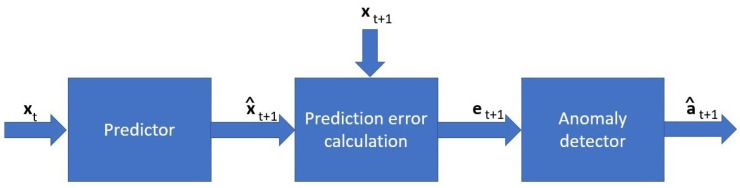
Process of anomaly detection.

**Figure 3 sensors-23-05013-f003:**
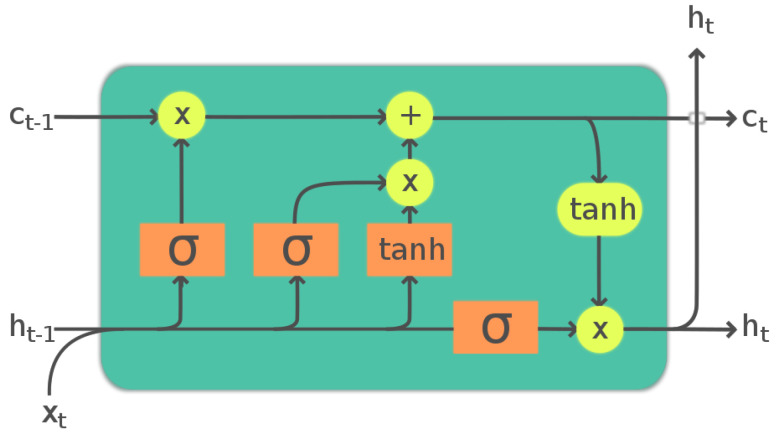
The Long Short Term Memory cell. $h_{t}$ and $c_{t}$ are the hidden and context vectors respectively. σ stands for the sigmoid activation function.

**Figure 4 sensors-23-05013-f004:**
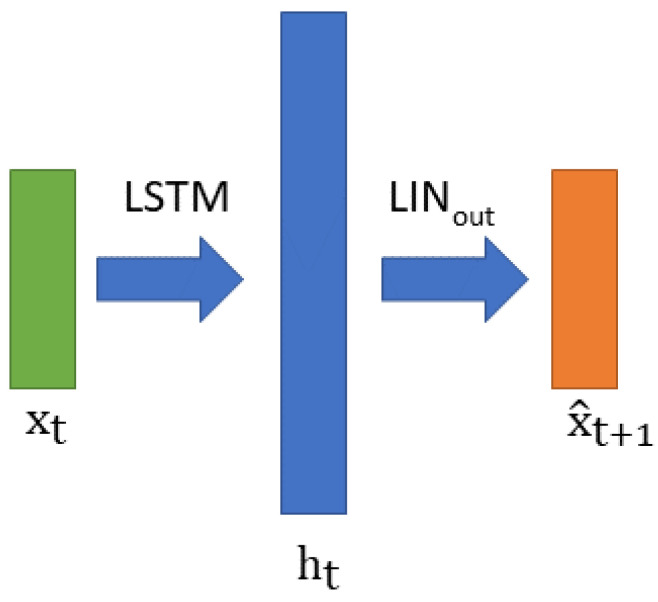
Organization/architecture of the Long Short Term Memory.

**Figure 5 sensors-23-05013-f005:**
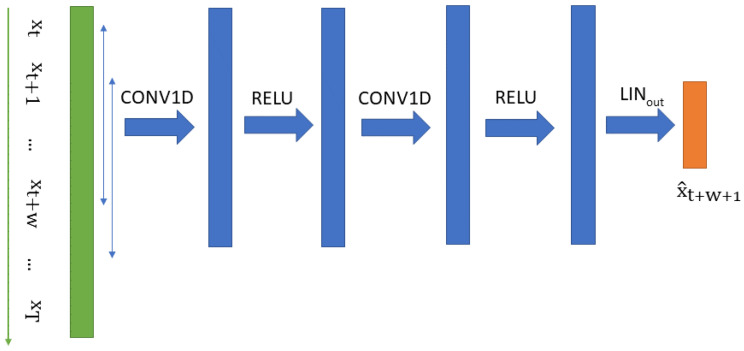
Organization/architecture of the Convolutional Neural Network.

**Figure 6 sensors-23-05013-f006:**
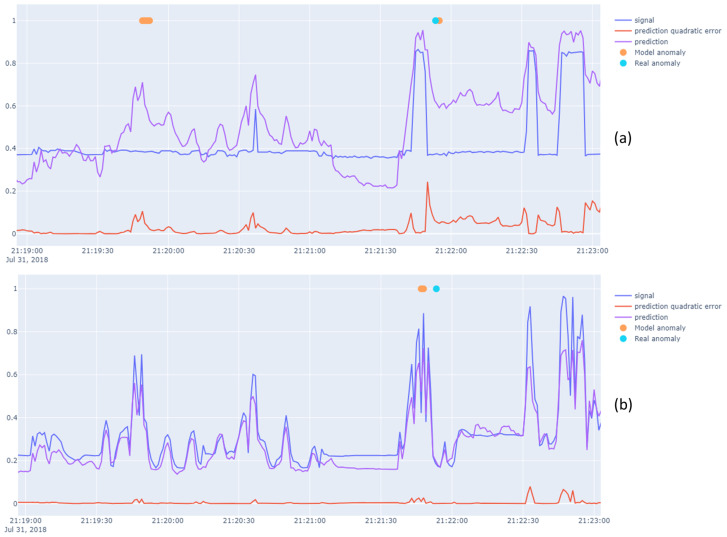
Blue: signal, Results for a slice of the engine oil pressure (**a**) and engine mean effective torque (**b**) time series. On the left side, we see a specific abnormal behavior of the oil pressure. As the mean effective torque rises, the oil pressure is supposed to do so as well, but it does not do so here, which constitutes an abnormal behavior that our model successfully highlighted despite a label is missing at this timestep. On the right side, we see an example of a true positive and the impact of the anomaly likelihood, which adapts to the variation of the prediction error from the left to the right side. This method also induces a little delay in the anomaly detection time because of the local average that is computed.

**Figure 7 sensors-23-05013-f007:**
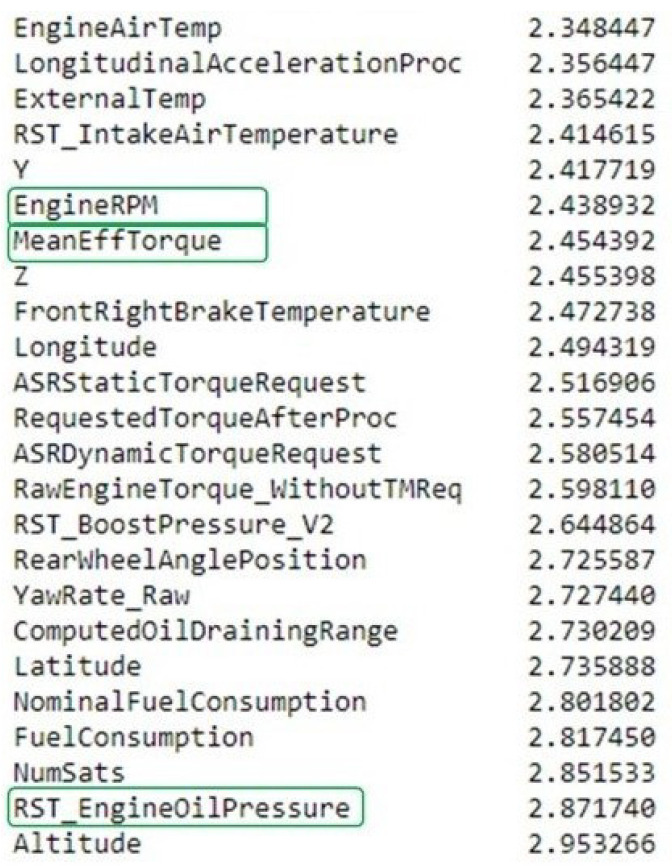
Variables correlations with labels. Variables are shown ordered by decreasing PLR. Circled in green are the variables that are known by experts to have caused the oil pressure failure.

**Table 1 sensors-23-05013-t001:** Comparison of state-of-the-art anomaly detectors on the public benchmark Yahoo [32]. The LSTM (ours) corresponds to the LSTM with 50 cells and two layers, as presented in Section 4, using the semi-supervised threshold. A1 to A4 correspond to different groups of time-series that can be found within the Yahoo benchmark.

Model	A1	A2	A3	A4	Mean
Hierarchical Temporal Memory (HTM) [33]	0.59	0.66	0.33	0.29	0.47
Online evolving Spiking Neural Network (OeSNN-UAD) [9]	0.70	0.69	0.41	0.34	0.54
Deep learning-based Anomaly detection approach for Time-series (DeepAnT) using a Long Short Term Memory (LSTM) [10]	0.44	0.97	0.72	0.59	0.68
Deep learning-based Anomaly detection approach for Time-series (DeepAnT) using a Convolutional Neural Network (CNN) [10]	0.46	0.94	0.87	0.68	0.74
Time-series Anomaly Detection using Generative Adversarial Networks (TadGAN) [13]	0.8	0.87	0.69	0.6	0.74
Long Short Term Memory (LSTM) (ours)	0.71	1	0.82	0.76	0.82

**Table 2 sensors-23-05013-t002:** Performance evaluation comparison between the LSTM [17], GRU, and CNN [10] models. CNN (n,m) refers to a CNN with n channels in the first layer and m channels in the second layer, while LSTM (n,m) and GRU (n,m) refer to an LSTM or GRU with n cells in the first layer and m cells in the second layer.

Model	$w_{a}$	f1	TPR	FPR	MSE	MACS	#Parameters
CNN (8-8)	5 s	0.114	0.254	$1.66\times 10^{-4}$	0.012	102,240	10,564
CNN (16-16)	5 s	0.114	0.254	$1.84\times 10^{-5}$	0.009	211,392	21,812
CNN (32-32)	5 s	0.12	0.253	0	0.007	450,432	46,612
**GRU (50-50)**	**5 s**	0.114	0.261	$3.12\times 10^{-5}$	0.003	**40,600**	**39,984**
GRU (100-100)	5 s	0.114	0.261	$2\times 10^{-5}$	0.002	126,200	124,884
**GRU (150-150)**	5 s	0.114	0.261	0	0.002	256,800	254,784
LSTM (50-50)	5 s	0.114	0.261	$1.34\times 10^{-5}$	0.005	52,600	51,884
LSTM (100-100)	5 s	0.114	0.261	$2.45\times 10^{-5}$	0.003	165,200	163,684
LSTM (150-150)	5 s	0.114	0.261	$2.67\times 10^{-5}$	0.003	337,800	335,484
CNN (8-8)	30 s	0.241	0.251	$5.44\times 10^{-5}$	0.012	102,240	10,564
CNN (16-16)	30 s	0.241	0.251	0	0.009	211,392	21,812
CNN (32-32)	30 s	0.247	0.249	0	0.007	450,432	46,612
GRU (50-50)	30 s	0.243	0.258	0	0.003	40,600	39,984
GRU (100-100)	30 s	0.243	0.258	$1.31\times 10^{-5}$	0.002	126,200	124,884
GRU (150-150)	30 s	0.243	0.258	0	0.002	256,800	254,784
LSTM (50-50)	30 s	0.243	0.258	$1.31\times 10^{-5}$	0.005	52,600	51,884
LSTM (100-100)	30 s	0.243	0.258	0	0.003	165,200	163,684
LSTM (150-150)	30 s	0.243	0.258	0	0.003	337,800	335,484
CNN (8-8)	60 s	0.265	0.249	$2.66\times 10^{-5}$	0.012	102,240	10,564
CNN (16-16)	60 s	0.265	0.249	0	0.009	211,392	21,812
**CNN (32-32)**	60 s	0.268	0.244	0	0.007	450,432	46,612
**GRU (50-50)**	60 s	0.267	0.254	0	0.003	**40,600**	**39,984**
GRU (100-100)	60 s	0.267	0.254	$2.58\times 10^{-5}$	0.002	126,200	124,884
GRU (150-150)	60 s	0.267	0.254	0	0.002	256,800	254,784
LSTM (50-50)	60 s	0.266	0.254	$2.58\times 10^{-5}$	0.005	52,600	51,884
LSTM (100-100)	60 s	0.266	0.254	0	0.003	165,200	163,684
LSTM (150-150)	60 s	0.266	0.254	0	0.003	337,800	335,484

**Table 3 sensors-23-05013-t003:** Performance evaluation of the LSTM (50-50) on limited labels.

$w_{a}$	TPR	FPR	TNR	FNR
5 s	0.19	0.065	0.91	0.0093
30 s	0.62	0.062	0.079	0.0082
60 s	0.86	0.061	0.64	0.0076

**Table 4 sensors-23-05013-t004:** Performance evaluation comparison between different Long Short Term Memories for $w_{a}=60$ s.

Model	TPR	FPR	TNR	FNR
LSTM (10)	0.86	0.061	0.64	0.0076
LSTM (50)	0.68	0.048	0.65	0.0081
LSTM (10-10)	0.84	0.068	0.63	0.008
LSTM (50-50)	0.69	0.049	0.65	0.008

**Table 5 sensors-23-05013-t005:** Computation costs comparison between different LSTMs; the PLR is computed with respect to $w_{a}=60$ s.

Model	Test MSE	PLR	MACS	#Parameters
LSTM (10)	0.0059	14.1	4.8 K	4.8 K
LSTM (50)	0.0021	14.2	31.8 K	31.5 K
LSTM (10-10)	0.007	12.4	5.7 K	5.6 K
LSTM (50-50)	0.002	14	52.6 K	51.9 K

## Data Availability

The Yahoo anomaly benchmark can be found here: https://webscope.sandbox.yahoo.com/catalog.php?datatype=s&did=70 (accessed on 12 Jaunary 2022). The Renault car data is publicly unavailable.

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
