# Peer review of "Unsupervised Anomaly Detection for Cars CAN Sensors Time Series Using Small Recurrent and Convolutional Neural Networks"

_sensors, 2023, doi:10.3390/s23115013_

Round 1
Reviewer 1 Report
· The paper's organization needs to be added in the Introduction section.
· Flow chart of the Proposed Work is missing.
· No of typos in the paper.
· The author must explain the GRU, LSTM and CNN algorithms in the proposed section.
· How these algorithms improve the efficiency of the work.
· What is the significance of Adam Optimizer in the result section? Justify it.
Author Response
Dear Madam, dear Sir,
Thank you for your review. We took care of each of your concerns as listed below. Please find attached the PDF of the revised paper. We have highlighted in blue all our modifications.
- The paper's organization needs to be added in the Introduction section.
We added a description of the paper organization in the introduction. (from line 123 of the revised version)
- Flow chart of the Proposed Work is missing.
We added a complete flow chart of our work in Figure 1.
- No of typos in the paper.
We corrected the found typos.
- The author must explain the GRU, LSTM and CNN algorithms in the proposed section
We added a description of these models and their architectures in section 3.2. (lines 207 to 236)
- How these algorithms improve the efficiency of the work.
While the LSTM and CNN are state-of-the-art algorithms that can be used to do unsupervised anomaly detection, the GRU can help to reduce the number of parameters and MACs (Multiply and ACCumulate) operations of the anomaly detector while giving similar or even better performance as shown in the paper. We added some quantified comparisons in section 4. (line 350)
- What is the significance of Adam Optimizer in the result section? Justify it.
The Adam optimizer is one of the most commonly used optimizer algorithms in the field of Machine Learning. We can find it in almost any Gradient Descent based work using RNNs or CNNs in the literature and constitutes a training milestone for these neural networks. We added this argument in section 4. (line 274)

Reviewer 2 Report
I would like to congratulate the authors on the experimental work carried out. Few of my concerns are provided below.
1. The Abstract should be well structured with quantified results.
2. The Introduction section must be improved with more recent research works. The novelty and contributions must be more effective portraying the research gaps.
3. Why did the authors chose LSTM ? There are numerous techniques available. The reason for selection is missing.
4. The overall architecture of LSTM is missing.
5. The authors state that they are using LSTM(50-50). For a layman reader, the understanding of LSTM(50-50) will not be conveyed. Also, LSTM (10-10), LSTM(50), LSTM(10). Kindly clarify.
6. Performance metrics description is missing.
7.The metric finesse is new.
8. Line 107, is it GO or GB?
9. What type of filter did the authors use? Dataset description is not clear. The authors say 486 variables were recorded. I am not able to understand the description.
10. State of the art comparison required.
Author Response
Dear Madam, dear Sir,
Thank you for your review. We took care of each of your concerns as listed below. Please find attached the PDF of the revised paper. We have highlighted in blue all our modifications.
- The Abstract should be well structured with quantified results.
We restructured it and added some quantified results.
- The Introduction section must be improved with more recent research works. The novelty and contributions must be more effective portraying the research gaps.
We added 16 recent references and a more detailed explanation of the state of the art in anomaly detection and clarified our contribution in the introduction.
- Why did the authors chose LSTM ? There are numerous techniques available. The reason for selection is missing.
We added justifications of our model selection in the introduction and in section 3 with state-of-the-art comparisons. (line 194)
- The overall architecture of LSTM is missing.
We added the LSTM architecture and cell description in section 3 and figure 3 and 4. Line(207)
- The authors state that they are using LSTM(50-50). For a layman reader, the understanding of LSTM(50-50) will not be conveyed. Also, LSTM (10-10), LSTM(50), LSTM(10). Kindly clarify.
We clarified this notation in each model’s description in section 3 lines (207, 226, 234) as well as below the table 5.
- Performance metrics description is missing.
We added detailed performance metrics description in section 4 equations 20 to 23.
- The metric finesse is new.
We found an existing name for it (Positive Likelihood Ratio (PLR)) and used it in section 4 (line 324).
- Line 107, is it GO or GB?
We corrected it to GB (line 154 now).
- What type of filter did the authors use? Dataset description is not clear. The authors say 486 variables were recorded. I am not able to understand the description.
We clarified the pre-processing and resulting data in section 2. (line 156)
- State of the art comparison required.
We added a state-of-the-art comparison of our selected models in table 1. This table is discussed in the section 3, from line 194.

Reviewer 3 Report
The authors proposed Unsupervised anomaly detection for cars CAN sensors time series using small RNN and CNN, but this study in unable to accept in current form, unless improve thoroughly:
It is better not to use abbreviation in the title, abstract and keywords.
Introduction part is missing the sufficient background and include all relevant
references in problem statement and describing the main idea.
The discussion part in not enough to discuss the different model results comparisons.
Author Response
Dear Madam, dear Sir,
Thank you for your review. We took care of each of your concerns as listed below. Please find attached the PDF of the revised paper. We have highlighted in blue all our modifications.
- It is better not to use abbreviation in the title, abstract and keywords.
We changed the abbreviations of LSTM, GRU and CNN but kept the CAN one as it is a technical generic abbreviation in the car industry that seems more recognizable as is.
- Introduction part is missing the sufficient background and include all relevant
references in problem statement and describing the main idea.
We added 16 recent references and a more detailed explanation of the state of the art in anomaly detection. We also clarified our contribution and the main problematic in the introduction.
- The discussion part in not enough to discuss the different model results comparisons.
We added a quantified comparison in section 4 (line 350) and developed the discussion.

Round 2
Reviewer 1 Report
The author has made all the suggested changes in the manuscript. Now this paper is acceptable, and we may proceed further.
Reviewer 2 Report
The authors have made significant changes to the manuscript. I recommend for publishing the article.